# Advances and Challenges Associated with Low-Cost Pulse Oximeters in Home Care Programs: A Review

**DOI:** 10.3390/s24196284

**Published:** 2024-09-28

**Authors:** Anisbed Naranjo Rojas, Freiser Cruz Mosquera

**Affiliations:** 1Health and Education Research, Group (GINEYSA), Department of Health, Universidad Santiago de Cali, Cali 760001, Colombia; 2Biomedicine Doctoral Program, Universidad de Córdoba, 14001 Córdoba, Spain; 3Integral Health Research Group (GISI), Department of Health, Universidad Santiago de Cali, Cali 760001, Colombia; freiser.cruz00@usc.edu.co

**Keywords:** pulse oximeters, home care, cost-effectiveness, remote monitoring, respiratory conditions, health technology

## Abstract

Oximeters have significantly evolved since their invention and are essential for monitoring chronic diseases in home care. However, commercial models can present an economic barrier. Therefore, we conducted a review of the use of low-cost pulse oximeters in the home care of patients with respiratory diseases. Our review included studies addressing oxygen saturation and heart rate monitoring in adults, focusing on the use of portable devices. Our search identified advances in vital signs monitoring that could provide accessible solutions for non-clinical settings. Although there are challenges related to clinical validation and accuracy, these oximeters may improve medical care, particularly in resource-limited areas. As a result, the accessibility of these devices opens up new possibilities for patients with chronic respiratory diseases in home care, enabling regular self-monitoring and increasing control over their health.

## 1. Introduction

Pulse oximeters have significantly evolved since their invention in the 1930s [1,2,3] to their integration into contemporary medical practice. They are a fundamental tool in home care for monitoring patients with chronic respiratory conditions [4,5,6,7]. However, commercial pulse oximeters can present an economic barrier for some patients participating in home care programs [8,9,10,11]. The World Health Organization emphasizes the need for accessible devices to improve the monitoring of chronic patients [12,13]. Additionally, previous research underscores the necessity of having affordable clinical monitoring equipment or tools [14,15,16], highlighting that economic constraints limit participation in these programs, especially in developing countries [17,18]. Therefore, there is a knowledge gap that necessitates the development and evaluation of a cost-effective and efficient pulse oximeter for home care monitoring [17,18,19].

Furthermore, different regions of the world face specific challenges related to access to home care and clinical monitoring devices [20,21]. In some rural or remote areas, inadequate infrastructure and limited medical resources hinder home care services and the availability of specialized medical devices [22,23,24], such as pulse oximeters. Additionally, in resource-limited countries, equitable access to home care can be difficult, leading to disparities in access to the necessary medical technology for monitoring patients with chronic illnesses [25,26,27]. Therefore, approaches are needed that can be adapted to the specific needs of each community and the availability of local resources [28,29] to address these regional challenges and ensure that all patients have access to adequate healthcare and clinical monitoring devices [30]. Strengthening home care, therefore, reflects an effort to democratize access to technology and improve quality of life, especially in resource-limited regions. These advances enable more people to benefit from technological innovations, promoting equity and social progress in various areas and globally [29,31].

The following provides a brief overview of the manuscript’s structure and the content covered in each section. The Section 2 examines the fundamental role and significance of pulse oximeters in home-based health monitoring. Section 3 discusses the development and specific challenges of affordable devices designed for home care settings. The Section 4 evaluates the advantages these devices offer in clinical environments and their impact on patient care. Lastly, Section 5 presents a detailed comparison between low-cost oximeters and commercial models, highlighting differences in performance, cost, and accessibility.

## 2. Pulse Oximeters in Home Monitoring

Pulse oximeters are critical in home care, as they allow continuous, non-invasive monitoring of blood oxygen saturation and heart rate. As shown in Figure 1, this process is based on a simple yet effective principle. Commercial oximeters operate by emitting red light (around 660 nm) and infrared light (around 940 nm) through a tissue, usually the finger. Oxyhemoglobin and deoxyhemoglobin absorb these wavelengths differently: oxyhemoglobin absorbs more infrared light and less red light, while deoxyhemoglobin absorbs more red light and less infrared light. A photodetector placed on the opposite side of the tissue measures the intensity of the light transmitted at both wavelengths [22]. Additionally, during measurement, the pulse oximeter emits these wavelengths through the tissue (typically the fingertip or earlobe), and a sensor detects the amount of light that passes through. By analyzing the ratio of absorbed red and infrared light, the device calculates the blood oxygen saturation level (SpO_2_). Additionally, the fluctuations in light absorption with each heartbeat allow the device to determine the patient’s heart rate. This process, based on pulse spectrophotometry, is fundamental to the operation of the oximeter and provides critical data for clinical monitoring [22,23].

On the other hand, low-cost pulse oximeters have gained relevance worldwide, especially in resource-limited countries. These devices allow accessible and continuous monitoring of oxygen saturation, which is vital for managing chronic diseases, preventing exacerbations, and adjusting treatments in a timely manner. The availability of affordable oximeters facilitates their use in both hospital and home settings, thus improving the quality of care and patient safety. Below are the most relevant chronic diseases that require continuous monitoring with pulse oximeters, particularly in respiratory diseases:

a. Chronic Obstructive Pulmonary Disease (COPD): It is one of the leading causes of morbidity and mortality worldwide. Patients with COPD often experience hypoxemia, making the use of pulse oximeters crucial for monitoring their oxygen saturation and adjusting home oxygen therapy [2,4].

b. Asthma: Patients with severe asthma may experience episodes of oxygen desaturation, especially during exacerbations. Monitoring with pulse oximeters helps identify these episodes and adjust treatment accordingly [5,6].

c. Cystic Fibrosis: This genetic disease affects the lungs and other organs. Patients often have recurrent lung infections and progressive deterioration of lung function, making oxygen saturation monitoring essential [7,9].

d. Pneumonia: In severe cases, especially in patients with other comorbidities, pneumonia can lead to a significant decrease in blood oxygen levels, justifying the use of pulse oximeters for continuous monitoring [9,10].

e. Sleep Apnea: Patients with this condition may experience nocturnal hypoxemia. Monitoring with pulse oximeters can help diagnose the severity of apnea and the effectiveness of CPAP (Continuous Positive Airway Pressure) treatment [11].

f. Heart Failure: Although not exclusively a respiratory disease, patients with heart failure may experience hypoxemia secondary to pulmonary edema. The use of pulse oximeters can be helpful in monitoring these patients [4,5,9].

Another important aspect to highlight is that these devices, by providing real-time data on oxygen levels, help to quickly detect any deterioration or favorable or unfavorable change in patients’ health, allowing for early intervention and reducing the risk of complications. Additionally, their ease of use and accessibility make them essential tools for self-care and health management at home, improving quality of life and reducing the likelihood of exacerbations and frequent rehospitalizations [4,8].

Moreover, these devices allow constant monitoring of blood oxygen saturation, which is important for detecting any drop in oxygen levels that may indicate a worsening of patients’ signs and symptoms [17,18]. The ability to quickly identify changes in oxygenation allows for the detection of warning signs, which, if not detected in time, can lead to hospitalizations or, in the worst cases, fatalities [28,29].

On the other hand, the design and use of low-cost pulse oximeters in home care offer a viable solution for continuous monitoring of patients with respiratory conditions. They also provide the possibility of increasing coverage for chronic patients who require these devices and obtaining comprehensive care that includes reliable monitoring tools for their respiratory condition [30,31]. These devices feature a design that does not compromise their functionality; on the contrary, it facilitates their use by patients without medical training, allowing for easy interpretation of results thanks to clear displays and audible or visual notifications in case of abnormal values [30]. The above demonstrates that the implementation of low-cost oximeters in the home is especially beneficial in resource-limited areas, as it reduces gaps in care [17,30].

## 3. Low-Cost Oximeters: Perspectives in Home Care

As previously mentioned, oximeters are especially beneficial for patients with chronic respiratory diseases, as they can perform regular monitoring without solely relying on medical visits, reducing travel that could lead to complications [4,6]. However, these devices also face significant challenges. Among them are potential variations in measurement accuracy due to the quality of the components used, as well as the need for adequate user education to correctly interpret the results and make health-related decisions. Additionally, the durability and long-term reliability of low-cost oximeters can be important issues to consider, as they may require periodic maintenance or calibration to ensure accurate and consistent measurements [18,20].

In this regard, one of the previously mentioned challenges faced by low-cost oximeters is the selection of components that are economical yet maintain high standards of accuracy and reliability in measurements [30]. This includes choosing optical sensors that can accurately detect variations in light absorption by oxygenated and deoxygenated hemoglobin at different wavelengths [29,30,31]. Additionally, the quality of microcontrollers and signal processing algorithms is crucial to ensuring accurate and consistent measurements [27,28]. Another significant challenge is the calibration and validation of these devices to ensure that the results obtained are reliable and comparable to established standards. Moreover, durability and resistance to adverse environmental conditions must be considered, especially for prolonged use in patients’ homes. Overcoming these technical and logistical challenges is essential to maximizing the benefits and effectiveness of low-cost oximeters in health monitoring in home settings [22,23,24].

In this same context, the design of low-cost oximeters also faces challenges in terms of ergonomics and usability. It is crucial to develop devices that are intuitive and easy to use for patients and caregivers in home settings [20]. This involves considering factors such as the size and weight of the device, the display, and the ease of interpretation of the results by the user. Additionally, energy autonomy is another important aspect, as many low-cost oximeters operate on batteries that must be efficient and long-lasting to ensure continuous use without interruptions [12,13]. The durability and resilience of the materials are also essential to ensure that these devices can withstand daily use and variable temperature and humidity conditions in homes. Addressing these challenges not only improves the user experience but also contributes to the widespread acceptance and adoption of low-cost oximeters as effective tools for health monitoring in home care [28,29,30,31]. Figure 2 illustrates the key challenges associated with the implementation of low-cost pulse oximeters in home care, highlighting the critical aspects that must be addressed to ensure the effectiveness and safety of these devices in patient monitoring at home.

Similarly, beyond the technical and usability challenges, the integration of low-cost oximeters into home care also faces challenges related to user education and training [24,25]. As mentioned earlier, it is important to provide adequate training to patients and caregivers on how to correctly use the device and interpret the results [18,20]. A lack of understanding or incorrect knowledge about how the oximeter works could lead to misinterpretations of oxygen saturation values, which could result in incorrect decisions regarding the patient’s health management [20,31]. Therefore, investing in continuous education programs and clear, accessible instructional materials is essential to maximize the benefits of low-cost oximeters in home care and enhance patient autonomy and self-care [30,31].

On the other hand, the development of low-cost pulse oximeters is an area of growing interest in medical technology, especially in the context of the increasing demand for accessible and accurate devices for health monitoring. To ensure that these devices are effective and reliable, it is crucial to consider various technical and methodological aspects. Table 1 outlines the aspects to consider for developing a low-cost oximeter, including signal processing, sensor types, and communication protocols that are essential for the design of low-cost pulse oximeters [32]. These elements not only guarantee the accuracy of the device but also optimize its functionality and accessibility, which is particularly important in resource-limited settings.

On the other hand, some authors [30,31,32] have highlighted the importance of implementing training strategies to improve the practices of healthcare providers in low- and middle-income countries. These strategies include in-service training, educational visits, and peer education, which are essential to ensure the proper use of medical devices and maximize their benefits. Community education and support also play a vital role in the acceptance and proper use of these devices by patients, especially in resource-limited settings. For example, designing community health programs accompanied by frequent training can help healthcare workers identify and manage medical conditions using low-cost technologies. This not only improves patient care but also reduces long-term healthcare costs by preventing serious complications [33].

Similarly, training and education are essential components for the successful implementation of low-cost medical devices. They ensure that both healthcare professionals and patients use these devices effectively, resulting in better health management and quality of life for underserved communities.

In this context, the implementation of low-cost pulse oximeters in home care programs could not only benefit individual patients [30,31] but also have a positive impact on healthcare systems [28,29]. By enabling continuous and accurate monitoring of vital signs, oximeters could help prevent serious complications [30,31], reduce unnecessary visits to healthcare facilities, and lower the costs associated with managing chronic diseases. Additionally, by facilitating the early detection of changes in patients’ health, low-cost pulse oximeters could contribute to a prevention-focused approach, thereby optimizing healthcare resource management and improving long-term health outcomes [27,28]. Therefore, this literature review aims to determine the potential use and benefits of low-cost oximeters for self-monitoring of vital signs among patients in home care programs.

Pulse oximeters in home care are evolving significantly, contributing to clinical monitoring and follow-up [32,33]. The reviewed studies highlight the important role that low-cost pulse oximeters could play in the monitoring and follow-up of patients with respiratory diseases [33,34]. In addition to measuring health status, they can predict clinical stability based on oxygen saturation and other vital signs, such as heart rate, and extend monitoring beyond conventional clinical settings.

Moreover, by expanding the scope of clinical monitoring, low-cost pulse oximeters empower patients to actively manage their health and promote holistic, patient-centered care [34,35]. In this regard, they are essential for improving health outcomes and quality of life in home care.

In the same context, the use of low-cost pulse oximeters in home care can significantly reduce complications associated with hospitalization [32,34]. By enabling continuous and non-invasive monitoring of blood oxygen saturation, these devices facilitate the early detection of changes in a patient’s health status and allow patients and caregivers to recognize important changes in disease behavior by understanding basic information about clinical signs [34,35]. They also allow for rapid intervention, which can prevent the progression of respiratory conditions to critical situations requiring hospitalization and exacerbations [31,32]. Additionally, by reducing the need for hospitalizations, the risk of nosocomial infections and other complications that can arise during a prolonged hospital stay is minimized [35]. In this way, pulse oximeters not only help improve the quality of life for patients by keeping them out of the hospital but also alleviate the burden on hospital resources and healthcare staff, creating a more efficient and safer healthcare system [34,35].

Similarly, integrating low-cost pulse oximeters into the treatment of chronic patients as part of routine home care would not only help reduce the burden on medical facilities and staff but also optimize resource allocation [34,35]. Additionally, by facilitating the early detection of complications in chronic patients, particularly those with respiratory conditions, low-cost pulse oximeters enable rapid interventions that can prevent severe complications, thereby avoiding costly hospitalizations. This preventive approach not only improves the quality of life for patients but also generates significant savings for healthcare systems by reducing expenses associated with advanced disease treatment and emergency care [31,32,33,34].

An important aspect to consider in the use of these devices in home care is patient education on the use of low-cost oximeters, which can have a significant impact on disease management in home settings [35]. By teaching patients and their caregivers how to use these devices correctly, they are empowered to monitor their oxygen saturation levels and heart rate accurately and regularly [32,33]. This knowledge enables them to detect any anomalies or deterioration in their health condition early, facilitating timely and appropriate interventions. This is closely related to the possibility of reducing potential complications and hospitalizations associated with a lack of knowledge about the disease and warning signs that indicate the need to consult a professional or seek emergency services [34].

Moreover, training in the use of pulse oximeters can foster a sense of responsibility and self-management in patients regarding their signs and disease, promoting a more active and participatory approach to managing their own health. This self-care can reduce the dependence on frequent medical visits and lessen the burden on healthcare services [33,35]. Education on the use of low-cost oximeters not only enhances patients’ ability to manage their health conditions but also optimizes treatment outcomes and the efficiency of the home care system [34].

On the other hand, low-cost oximeters prove to be viable for home care, as the materials used in their design allow for more affordable manufacturing without significantly sacrificing accuracy and functionality [29,30,31,32]. These devices often employ economical but efficient optical sensors and microcontrollers, along with durable plastic housings, which reduce production costs compared to conventional oximeters that may use more expensive components and sophisticated materials [34,35]

Although commercial oximeters may offer greater durability and accuracy in certain clinical contexts, low-cost oximeters have proven to be adequately accurate for regular home monitoring [35]. This economic accessibility allows for wider distribution, especially in resource-limited communities, thus democratizing access to essential health monitoring tools [32,33,34,35]. On the other hand, an interesting aspect to address in the context of low-cost oximeters in home care is interoperability and digital connectivity. Currently, the ability of these devices to integrate with electronic health systems and telemedicine platforms represents a significant opportunity to improve remote health management [35,36]. This interoperability would allow healthcare professionals to access real-time data, monitor patients more effectively, and offer more personalized and timely interventions from any location [32,33,34,35]. Additionally, digital connectivity would facilitate continuous patient education on the proper use of the oximeter and the interpretation of results, thereby promoting greater empowerment and autonomy in home health self-care. Integrating these capabilities could not only overcome some of the current limitations of low-cost oximeters but also significantly expand their utility and effectiveness in modern home settings [34,35].

In this same vein, it is relevant to explore the relationship between the interoperability and connectivity of low-cost oximeters and the efficiency and quality of the healthcare system [36,37,38,39]. The integration of these devices with digital platforms could optimize the collection and analysis of health data, providing valuable information for managing chronic diseases and planning preventive interventions [30,31,32,33,34,35]. This could not only reduce the time and resources dedicated to in-person medical visits but also enhance the capacity of healthcare systems to proactively respond to changes in patients’ conditions [28,29]. Moreover, by facilitating continuous and remote monitoring, connected oximeters could contribute to more personalized and data-driven care, thereby improving health outcomes and satisfaction for both patients and healthcare providers. Ultimately, the effective implementation of digital connectivity in low-cost oximeters could promote a more efficient, accessible, and patient-centered healthcare system [34,35].

In the context of low-cost oximeters in home care, it is also interesting to address the durability and lifespan of these devices [31,32]. Medical devices used in home settings often face additional challenges due to factors such as continuous use, handling by un-trained patients, and variable environmental conditions [33,34]. Therefore, researching and improving the durability of low-cost oximeters could enhance their long-term effectiveness and reduce costs associated with frequent equipment replacement [33,34]. Additionally, ensuring that these devices can withstand adverse conditions typical of home environments, such as humidity or temperature fluctuations, is crucial for maintaining their accuracy and reliability. Improving durability would not only benefit patients by providing more consistent continuous monitoring but also strengthen the economic viability and sustainability of implementing low-cost oximeters in home care [33,34,35].

Thus, the successful integration of low-cost oximeters in home care requires a com-prehensive approach that encompasses everything from improving durability and interoperability to implementing effective educational programs [30,31,32,33,34]. Strengthening these key aspects will not only enhance the accuracy and usefulness of these devices in continuous health monitoring but also empower patients to take an active role in man-aging their well-being [35]. This benefits individuals by facilitating more proactive and personalized care, while also promoting greater efficiency and sustainability in healthcare systems, thereby optimizing available resources for more effective and patient-centered care [30,31,32]. As mentioned earlier, these devices become a viable and valuable option for improving health management at home [33,34].

Low-cost pulse oximeters have emerged as devices that provide accurate monitoring of blood oxygen saturation, especially in home care settings. This technology offers an accessible and convenient way for patients and healthcare providers to continuously monitor respiratory function and cardiovascular health. However, their implementation poses significant challenges that must be addressed with careful and meticulous evaluation. These challenges include the accuracy and reliability of the data collected, proper calibration of the device, necessary training for correct use by users, and appropriate interpretation of the results obtained. It is crucial to establish clear protocols and conduct rigorous studies to ensure that low-cost oximeters meet the quality and safety standards required in home care and other clinical settings.

## 4. Benefits of Low-Cost Pulse Oximeters in Clinical Monitoring

The development of a low-cost device for monitoring heart rate and blood oxygen saturation is of utmost importance for several reasons. Firstly, it addresses the growing need to monitor these vital signs in both clinical and home settings, offering a more accessible and practical solution [36,37]. This approach highlights its utility as a non-invasive and safe method for detecting vital signs. Additionally, the device’s ability to alert users when abnormal values are detected is crucial for preventing potential health issues. Similarly, the design proposed by Dai et al. [36] not only provides a tool for monitoring vital signs but also addresses concerns related to accessibility, convenience, and efficiency in medical care, emphasizing the importance of preventive care and self-care [38,39].

Moreover, the oximeter system designed by Altayeb et al. [40] represents a significant advancement in the efficiency and affordability of devices for monitoring heart rate and oxygen saturation. Its focus on the reuse of discarded probes should be highlighted, as it not only drastically reduces costs by using recycled sensors but also demonstrates an excellent cost–benefit ratio by achieving a low total cost compared to commercial devices. Additionally, the combination of affordable hardware and software makes this design highly adaptable and a promising tool for both hospital and home settings.

Furthermore, Nemomssa and Raj et al. [41] present a new low-cost pulse oximeter design that stands out for its portability and the possibility of being powered by a rechargeable battery or a smartphone, which expands its accessibility, especially in resource-limited settings. This combination of precision, ease of transport, and reduced cost has considerable potential to improve healthcare [29], particularly in clinical settings with limited resources.

In this context, the article by Bhuyan et al. [39] highlights three key aspects in the design and application of low-cost pulse oximeters. First, it emphasizes the accuracy of the device in measuring blood oxygen saturation (SpO_2_) and heart rate (HR), providing essential data for the early detection of health problems. Second, it underscores the use of Arduino, a versatile microcontroller known for its data processing capabilities and adaptability to various applications.

However, the most notable aspect is the cost-effective implementation of the system, which utilizes affordable components, thereby facilitating its mass deployment and accessibility for a broad range of users. The device’s effectiveness and accuracy in experimental measurements, when benchmarked against existing standards, further underscore the significance and viability of this technology. Collectively, these elements illustrate how advancements in low-cost pulse oximeters have the potential to transform health monitoring, making it more accessible and effective.

Table 2 below presents a detailed comparison of the studies conducted by some authors, highlighting how each approach addresses the challenges of developing affordable, reliable, and accessible pulse oximeters for use across various healthcare settings.

On the other hand, monitoring tools have been developed that could benefit the follow-up of low-income patients. A notable example is the research conducted by Nicholas Boyd et al. [42], in which they developed a monitoring probe for oxygen saturation. This study evaluated the usability of the tool in both low-resource and high-resource settings. The results showed that the probe provided reliable readings of blood oxygen saturation in less than a minute. Additionally, it was evident that similar tools available on the market have significantly higher costs, highlighting the importance of developing affordable and accessible solutions for monitoring in resource-limited environments.

Moreover, the development of these low-cost tools is crucial for improving healthcare in underserved communities. The ability to obtain quick and accurate readings of blood oxygen saturation can make a significant difference in managing respiratory diseases and other critical conditions. Boyd and his team’s research [42] not only demonstrates the technical feasibility of these affordable solutions but also their potential for widespread implementation in resource-limited healthcare systems. This accessibility can help reduce disparities in medical care, allowing more patients to receive proper and timely monitoring without incurring high costs. Thus, innovation in affordable and efficient medical devices is an essential step toward achieving global health equity.

Similarly, research conducted by Carina King et al. [43] observed that the design of a reusable probe for measuring oxygen saturation could offer reliable and high-performance assessments in low-resource settings. This finding highlights the crucial need to design accessible and effective medical devices that can provide rapid and accurate monitoring of patients in environments where resources are scarce. This underscores the necessity of obtaining reliable and rapid measurements as a fundamental part of effective medical condition management, ensuring that even in challenging settings, patients receive the necessary care in a timely and efficient manner. This approach not only promotes the accessibility of medical technology but also emphasizes its positive impact on the quality of life and treatment of respiratory diseases and other medical conditions in home settings.

In this context, Russell G.B. et al. [44] found that the costs associated with monitoring procedures using pulse oximeters could be significant in their work environment. They proposed laminating pulse oximeter sensors and inserting them into a disposable protective shield as a solution to reduce these costs without compromising response time or sensor accuracy. Their results demonstrated that laminating and using a protective shield on disposable sensors did not affect blood oxygen saturation measurements.

Moreover, the proposal by Russell G.B. et al. [44] not only aims at cost reduction but also at improving sustainability and efficiency in the use of medical devices. Laminating pulse oximeter sensors and protecting them with a disposable shield could not only economically benefit healthcare settings but also extend the lifespan of the sensors, thereby reducing waste generation. This innovative approach does not compromise measurement accuracy and could be especially beneficial in contexts where resources are limited or where more sustainable practices in medical device management are sought. Additionally, the validation of these techniques opens the door to future research to explore other potential applications of lamination and the use of shields in disposable medical devices, promoting progress toward more efficient and environmentally responsible practices in healthcare.

## 5. Comparison between Low-Cost and Commercial Pulse Oximeters

When comparing the effectiveness of low-cost pulse oximeters with traditional ones, some studies have found significant similarities in measurements [45]. For instance, in a study [45] that evaluated the accuracy of six low-cost pulse oximeters during the stability of arterial oxygen saturation between 70% and 100% in 22 healthy individuals, it was observed that several of these devices showed results comparable to more expensive units available on the market. The researchers concluded that low-cost pulse oximeters can perform equivalently to more expensive models. However, they suggested the need to continue developing affordable oximeters for clinical application.

In addition to the similarities found in accuracy between low-cost and traditional oximeters, it is crucial to emphasize the importance of improving reliability and consistency under various clinical conditions. For example, the ability of these devices to provide accurate measurements during lower oxygen saturation ranges, as well as their robustness against variations in peripheral perfusion and other environmental factors, remains an area of interest for future research. This approach not only seeks to validate their utility in diverse clinical settings but also to explore improvements in the accessibility and overall performance of low-cost pulse oximeters, thereby promoting their broader application in global healthcare.

In this context, comparing the accuracy of low-cost oximeters with commercial models is actively exploring how these devices can integrate new technologies to enhance their clinical utility. As such, some studies [44,45] are investigating the incorporation of additional sensors and advanced algorithms that could allow for more precise and personalized monitoring of oxygen saturation. These developments have the potential to facilitate the early detection of changes in patient health and improve the management of respiratory diseases and other critical conditions.

Similarly, research conducted by Kovesi T. et al. [46] identified that the use of a low-cost oximeter for adults did not show a clear trend in errors within the evaluated saturation range, which was 87–99%. They also observed that the low-cost pulse oximeter for adults performed quite well in larger children. The authors suggest that given the large number of devices available online and the constantly evolving technology, it remains necessary to conduct research to evaluate non-clinical oximeters.

In this regard, the results of this research highlight the importance of validating the effectiveness of low-cost oximeters in different populations and clinical contexts. It is important to consider factors such as age, weight, and specific health conditions when evaluating the accuracy and reliability of these devices. The variability in patient characteristics can significantly influence the performance of oximeters, underscoring the need for ongoing and detailed studies. At the same time, the accessibility of these tools could significantly improve health monitoring in resource-limited communities, provided, as previously mentioned, that their accuracy and consistency are ensured through rigorous and up-to-date research.

On the other hand, in a study conducted by Berkenbosch Y. et al. [47], oxygen saturation measurements were taken using a device designed by the authors alongside conventional pulse oximeters. The results showed that the measurements obtained with the designed tool estimated oxygen saturation with the same accuracy as those taken with conventional new-generation digital sensors. As mentioned earlier, these findings highlight the capability and importance of devices specifically designed to compete with traditional pulse oximeter technologies. The validation of new tools not only opens the door to more accessible and possibly more cost-effective options but also drives innovation in the field of health monitoring. It is vital that future research continues to evaluate these devices in a variety of clinical conditions and diverse populations to ensure their reliability and effectiveness. By doing so, access to accurate and affordable health monitoring can be significantly improved, especially in economically disadvantaged regions.

Thus, the authors [46,47] reflect the critical need for continuous and detailed validation of low-cost pulse oximeters. Technology and design methods are rapidly evolving, and it is vital that these devices are tested in different clinical conditions and in populations where the results can be generalized to ensure their effectiveness. If low-cost pulse oximeters prove to be accurate and consistent on a large scale, they could revolutionize healthcare, particularly in home settings. However, it is essential that these tools undergo rigorous testing and that their limitations are clearly understood and addressed, considering both patients and professionals. Figure 3 provides an overview of the main critical factors in the implementation of low-cost pulse oximeters in home care. Key aspects such as device accuracy, user training, economic accessibility, and regulatory considerations are highlighted, all of which must be addressed to ensure their effectiveness and safety in monitoring patients at home.

In this same context, a study conducted by Nemomssa H.D. et al. [48] developed a low-cost pulse oximeter powered by a smartphone. They performed measurements on 15 individuals, resulting in higher accuracy in oxygen saturation measurement compared to other existing low-cost pulse oximeters. This demonstrates that as these devices become more accurate and user-friendly, their acceptance and adoption by patients requiring frequent monitoring increase significantly. This is particularly relevant in home care, where the need for reliable and affordable devices for home monitoring is complex in some situations. The integration of advanced technologies, such as smartphone power, not only makes the devices more accessible but also allows for more continuous and personalized patient health monitoring.

Conversely, the research conducted by Chan C. et al. [49] designed a low-cost smartphone-based pulse oximeter, which was used in both patients with chronic obstructive pulmonary disease (COPD) and healthy individuals. They observed that, during exercise, patients with COPD needed to take pauses to ensure greater accuracy in measuring oxygen saturation. This finding once again demonstrates the importance of considering patients’ specific conditions when using low-cost monitoring devices. Referring to the research conducted by Nemomssa H.D. et al. [48], their results show high accuracy in measuring oxygen saturation under controlled conditions, while the study by Chan C. et al. [49] highlights the additional limitations and challenges that arise in more dynamic clinical contexts. The need for pauses during exercise for COPD patients indicates that while low-cost smartphone-powered pulse oximeters offer an accessible solution, their effective application may require specific adjustments based on individual patient needs.

In this regard, the results of both studies further reinforce the importance of validating these devices across a variety of conditions and populations before their widespread implementation, as previously mentioned [45,47]. Accurate oxygen saturation measurement is crucial for the proper management of chronic diseases such as COPD, and any monitoring device must be adaptable to the conditions of use. Smartphone-powered pulse oximeter technology has the potential to revolutionize home care, providing a valuable tool for continuous health monitoring. However, it is essential that these tools undergo rigorous testing and that their limitations are clearly understood and addressed. Additionally, it is important to continue strengthening the ongoing development of these technologies to improve their accuracy and ease of use. Future research could focus on optimizing the algorithms used by these devices to better handle variability in exercise and other physical activities. It is also crucial to explore ways to educate users, especially patients with chronic diseases, on how to correctly use these devices to obtain reliable results [44,47,49].

Moreover, it is important for medical technology developers to work closely with the medical community to identify and resolve any issues that may arise in the practical use of these devices [50]. A key point to consider in future research is that it should focus not only on the accuracy of measurements but also on the usability and accessibility of these devices for both patients and healthcare professionals [50]. Proper education and training on the use of these devices are also crucial to ensuring their effective and safe use.

In this regard, low-cost oximeters present both advantages and disadvantages compared to affordable commercial oximeters. Among the advantages are their economic accessibility, which is especially important in resource-limited regions and for low-income individuals [51,52]. These devices can provide basic and continuous monitoring of oxygen saturation and heart rate, being useful in situations where medical resources are scarce. However, the disadvantages include lower accuracy and reliability in measurements, as they may not be subject to the same rigorous calibration and testing standards as commercial oximeters. Additionally, low-cost oximeters may lack advanced features present in more expensive models, such as the ability to store historical data or connectivity with other digital health platforms. Table 3 presents a comparison between low-cost and commercial oximeters.

In comparing these characteristics, the limitations and specific functions, along with the evaluated conditions, are clearly highlighted when several data tables are consolidated. This consolidation provides an overall view of how low-cost oximeters stand against commercial models in terms of cost–benefit and clinical applicability, especially in resource-limited settings where accessibility and simplicity are crucial [52].

Similarly, low-cost oximeters may present significant limitations in oxygen measurement accuracy due to skin pigmentation. These devices, which use light sensors to measure blood oxygen saturation, may be less accurate in individuals with darker skin tones. This is because skin pigmentation can absorb some of the light emitted by the sensors, affecting the detection of reflected light and, consequently, the measurement of oxygen levels [53]. While some advanced commercial oximeters have been calibrated to minimize this issue, low-cost models often lack these adaptations, which can lead to inaccurate and potentially dangerous results for clinical decision-making in patients with different skin tones [54].

In this regard, pulse oximeters must comply with international standards to ensure their essential performance and accuracy. The ISO 80601-2-61:2011 standard sets the safety and performance requirements for these devices, including aspects such as the accuracy of oxygen saturation and pulse rate measurements. On the other hand, the ISO 17025:2017 standard [55] refers to the general requirements for the competence of testing and calibration laboratories, which are relevant for the calibration of pulse oximeters, ensuring that measurements are accurate and traceable [32,55].

Regarding the lifespan of low-cost pulse oximeters, this is a crucial factor to consider. These devices, while more economically accessible, may have a shorter lifespan due to lower-quality materials and less durable electronic components [56]. Regular calibration and maintenance, which can be costly or difficult to perform for low-cost devices, also affect their accuracy over time. Therefore, while low-cost oximeters are valuable for expanding access to SpO_2_ measurement, it is important to consider these limitations and conduct periodic quality checks to ensure their reliable performance.

In light of these considerations, this review makes several unique contributions to the field of low-cost pulse oximetry. First, it identifies critical knowledge gaps in current research, particularly in understanding the long-term reliability and accuracy of low-cost devices compared to their commercial counterparts. By highlighting these gaps, this review paves the way for targeted investigations into the durability and performance consistency of affordable oximeters. Additionally, the review proposes several future research directions, including the development of advanced calibration methods, integration of novel sensor technologies, and exploration of machine learning techniques to enhance device accuracy and user experience. Areas that remain underexplored, such as the impact of environmental variables on device performance and user feedback on practical usability, are also identified as promising fields for future studies. Addressing these areas could lead to significant advancements in the design and implementation of low-cost pulse oximeters, ultimately improving accessibility and effectiveness in diverse healthcare settings. Conversely, low-cost pulse oximeters, which utilize simple components and accessible techniques, are primarily designed for emergency situations or educational purposes. This approach offers an economical alternative in contexts where monitoring oxygen saturation is critical but commercial equipment may not be available. Although these devices are not recommended for continuous or clinical monitoring due to potential limitations in accuracy and reliability, they can be valuable in emergency scenarios requiring rapid assessment or in educational settings to demonstrate basic measurement principles.

Finally, the literature provides a comprehensive perspective on the benefits and challenges associated with low-cost pulse oximeters. These devices have great potential to improve health monitoring at home, but their success will depend on continuous innovation and validation in different clinical contexts. Despite the obstacles mentioned, the potential of these devices to optimize health management in home settings remains promising, as long as the relevant technical and educational aspects are effectively managed.

## 6. Conclusions

The availability of low-cost pulse oximeters opens new avenues for home care, particularly for patients with chronic respiratory diseases, as the ability to monitor their health from home can make a significant difference [51,52]. However, it is essential to address challenges such as clinical validation to ensure the accuracy and reliability of the data collected by these devices [53]. Additionally, proper training must be provided to ensure that patients and caregivers receive the necessary instructions on how to use these devices and interpret the results [52,53].

Similarly, the use of accessible devices for managing respiratory diseases presents an opportunity to enhance patients’ quality of life and improve the effectiveness of home care. Regular and accurate monitoring of vital signs empowers patients to take greater control of their health and serves as a valuable resource for healthcare professionals when making clinical decisions.

## Figures and Tables

**Figure 1 sensors-24-06284-f001:**
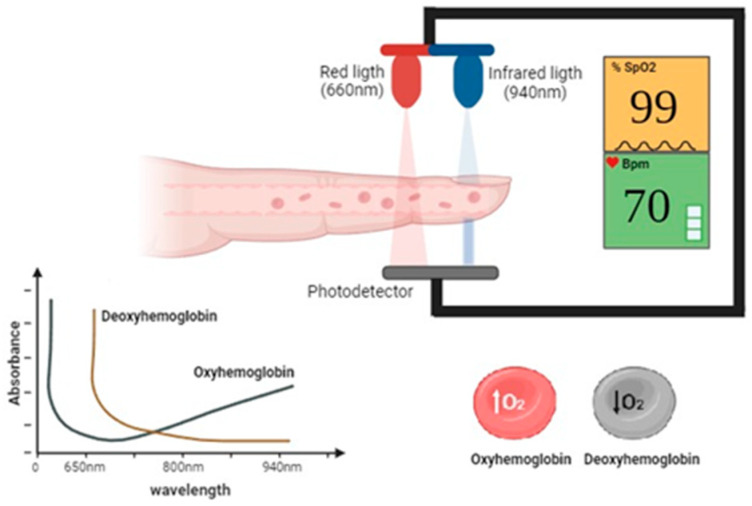
Operation of commercial pulse oximeters.

**Figure 2 sensors-24-06284-f002:**
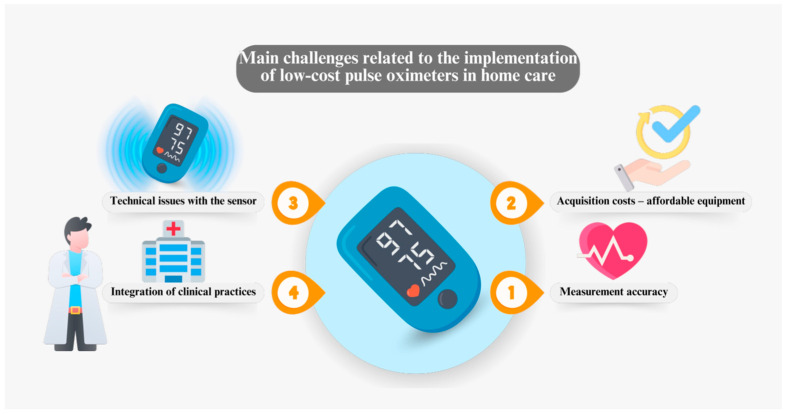
Main challenges related to the implementation of low-cost pulse oximeters in home care.

**Figure 3 sensors-24-06284-f003:**
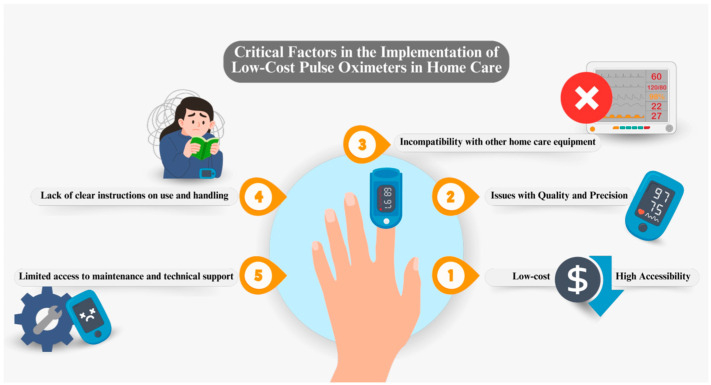
Critical factors in the implementation of low-cost pulse oximeters in home care.

**Table 1 sensors-24-06284-t001:** Key aspects to consider for developing a low-cost oximeter.

Aspect	Description
**Calibration Methods and Standards**	- Use established protocols for ensuring the safety and performance of oximeters [32]. - Calibrate with known reference samples and oxygen simulators.
**Signal Processing**	- Application of algorithms such as Kalman filtering, peak detection, and Fast Fourier Transform (FFT) analysis for noise and artifact elimination.
**Types and Characteristics of Sensors**	- Low-cost photodiode and infrared and red LED sensors. - Preference for sensors with high sensitivity and low crosstalk.
	- Use of I2C or UART protocols for sensor communication with microcontrollers like

**Table 2 sensors-24-06284-t002:** Comparison of Key Aspects in the Development of Affordable Pulse Oximeters by Various Authors.

Author	Objective	Main Components	Functions	Advantages	Clinical Applications
**Dai et al. [36]**	Design a portable and economical device to measure vital signs.	Arduino UNO, MAX30100 sensor, DS18B20 sensor.	Monitor heart rate, oxygen saturation, and body temperature.	Portable, displays data on an LCD screen.	Vital signs monitoring in home, clinical, and community settings.
**Altayeb et al. [40]**	Reduce costs by using recycled probes.	Recycled probe, Arduino microcontroller.	Measure and display heart rate and oxygen saturation.	Low cost, probe reuse, adaptable design.	Ideal for hospital settings and remote patient monitoring.
**Nemomssa and Raj [41]**	Improve accessibility in resource-limited settings.	Arduino-based system, smartphone-compatible.	High accuracy in measuring oxygen saturation and heart rate.	Economical, portable, high precision.	Useful for critical care, anesthesia, pre/post-surgery, especially during COVID-19.
**Bhuyan et al. [39]**	Implement an economical, low-power system for home monitoring.	Arduino microcontroller, affordable components.	Detect and display heart rate, oxygen saturation, and temperature.	Economical, easy to use, ideal for home monitoring.	Home-based health tracking for early detection of issues.

**Table 3 sensors-24-06284-t003:** Comparison of low-cost and commercial oximeters.

Aspect	Low-Cost Oximeters	Commercial Oximeters
**Sensor Type**	Simplified sensors (e.g., basic photodiodes or LEDs)	Advanced sensors (e.g., integrated phototransistors)
**Characteristics**	Economical components, basic technology	Advanced sensors, sophisticated algorithms
**Advantages**	Affordable, easy to use	High accuracy, clinically validated
**Disadvantages**	Lower accuracy, potential lack of rigorous validation	Higher cost, greater technological complexity

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
