# Peer review of "Advances and Challenges Associated with Low-Cost Pulse Oximeters in Home Care Programs: A Review"

_sensors, 2024, doi:10.3390/s24196284_

Round 1

Reviewer 1 Report

Comments and Suggestions for Authors

Abstract: The abstract states that this review only explores devices using Arduino technology. Why were other microcontrollers not considered? This is crucial due to signal sampling and processing accuracy.

Line 46: The relevance of quality of life and health in Colombia is mentioned. Why only this country? A broader global context should be included, highlighting the development of low-cost devices worldwide.

Section Pulse Oximeters in Home Monitoring: Include or list the most relevant chronic diseases or conditions requiring continuous monitoring with pulse oximeters, especially respiratory diseases or infections. The review should provide an overview and context for low-cost oximeters, prioritizing the most relevant diseases.

Line 74: A study is cited; it is necessary to include figures illustrating the simplicity of the design referenced.

If the review emphasizes advancements in low-cost oximeters and economic barriers to acquiring oximeters for certain populations, it should delve into a comparison of low-cost technology development against conventional devices. Compare costs, limitations, advantages, and highlight the requirements for proper validation of low-cost oximeters in section 4.2.

Low-Cost Oximeters Perspectives in Home Care: Include a table and figures on aspects to consider for developing a low-cost oximeter:

  • Calibration methods and current standards for medical device calibration (pulse oximeters).
  • Signal processing (mention the best algorithms).
  • Suitable sensor types and characteristics (used in previous literature on low-cost oximeters, including figures of commonly used sensors).
  • Sensor communication protocols with the Arduino microcontroller.

There should be a dedicated section on the challenges and limitations of low-cost pulse oximeters. This should include comparisons with commercial oximeters, citing previous studies and referencing additional literature: 10.1109/JSEN.2023.3235977

Include studies related to the use of low-cost health care devices demonstrating the importance of training and education to improve quality of life, particularly for low-resource populations.

Introduction section: The introduction should end with a brief outline of the review's structure and sections.

Materials and Methods: This section is not appropriate for a review. Authors should focus on summarizing the literature on low-cost pulse oximeters, their advantages, limitations, and applications in home care.

Section 2.1.1 is repeated twice.

If WiFi technology is mentioned, a section on advances in IoT pulse oximeters and integration with smartphones should be explored. I recommend these references to improve the manuscript quality:

·      https://doi.org/10.3390/electronics11193061

·      https://doi.org/10.3390/mi12080918

·      https://doi.org/10.3390/geriatrics7020043

·      https://doi.org/10.3390/s24113301

·      https://doi.org/10.1016/j.ohx.2022.e00309

·      https://doi.org/10.1007/s11277-019-06995-7

Results section: The first paragraph is redundant as it repeats the introduction’s points. Results should include tables or figures explaining the challenges of using low-cost pulse oximeters, citing previous studies. The mention of Table 1 and Figure 1 is incorrect; Figure 1 is not suitable for the review. 

Discussion section: The discussion should delve deeper into the advantages and disadvantages of low-cost oximeters compared to affordable commercial oximeters. In section 4.1, consolidate multiple tables into one, comparing limitations, costs, functions, and diseases tested with low-cost oximeters, especially in comparison with commercial ones.

Limitations in the accuracy of pulse oximeter measurements, such as skin pigmentation, should be mentioned in the manuscript: https://doi.org/10.1088/1361-6579/acd51a

Line 523: If low-cost oximeters were tested in specific populations with certain diseases, list them in a table for better clarity.

Elaborate on current standards for the essential performance of pulse oximeters, such as ISO 80601-2-61:2011 and ISO 17025:2017 for calibration. Discuss the lifespan of low-cost pulse oximeters, which is crucial for such devices. Also, detail calibration/signal processing for reliable and accurate results.

Include relevant citations to substantiate the conclusions.

Improve the writing style in the Acknowledgments section.

Include more figures and diagrams related to these devices, such as flowcharts of their functionality or block diagrams. Ensure tables are well-structured according to the journal’s format.

My recommendation:

Consider this work https://www.mdpi.com/2079-9292/9/11/1905, highlighting validation of measurements using Arduino ESP32 technology to improve the manuscript quality.

Comments on the Quality of English Language

Improve the quality of English, avoid repetition, and ensure the correct use of scientific language.

Reviewer 2 Report

Comments and Suggestions for Authors

Dear Authors,

Thank you for submitting your article "Advances and Challenges Associated with Low-Cost Pulse Oximeters in Home Care Programs: A Review" for review. The topic you are working on is very interesting and has the potential to connect knowledge in the field. I appreciate your enthusiasm for the issue, but it is necessary to approach it critically and consider its practical application.

However, in its current form, the processing of the article is insufficient. Let me highlight a few key areas for improvement:

I recommend a more thorough breakdown of the principle of operation and specific types of sensors and pulse oximeters that can be used or that were used by the authors of the reviewed articles. Mention their features, advantages and disadvantages - for example with a summary table. This section should be more detailed and structured. So far, there are only mentions in the article without mutual comparison.

The graphic materials are at a very low level and do not provide sufficient information. I recommend adding high-quality and informative pictures, diagrams and diagrams that would better illustrate your findings. The only diagram in the entire article showing the selection of articles for further analysis also needs to be graphically improved.

The Material and Methods section is confusing, the authors try to define what they want to explore in their review, but it is chaotic and seems like an effort to fit into some article template. Subsection 2.2 and 2.3 have the same title.

Sections Results and Discussion are very difficult to navigate and it will be very difficult for readers to draw connections. The combination of different texts and tables with a lot of text is difficult to understand.

I recommend accepting the article only after a thorough revision. In its current form, it is not suitable for publication, it needs to be finished. You need to work on the clarity, clarity and depth of the information you provide.

I hope that my comments will help you to improve your work. I wish you much success in the further processing of the article.

Reviewer 3 Report

Comments and Suggestions for Authors

This paper is a review paper on advances and challenges for oximeters, particularly low cost in for respiratory patients at home.

I believe the paper needs to be address in the following areas:

-            Misses description of the fundamental workings of an oximeter, specific light frequencies and how they are used to give an estimation of oxygen saturation and heart rate

-            Motivation for the paper needs to strengthened, cost is referred to without qualification.

o   Also, unclear what conventional is.

o   These devices can be bought online by patients easily for similar cost as Arduino breakout board

o   Title is one of the few places that reason for the review paper is given, to find challenges in the space for future research.

-            Incredibly restrictive inclusion and exclusion criteria with minimal justification for why the criteria were chosen

-            Also, how was a rapid review chosen as preferred approach, e.g,. what rate of publications in the field, and if high, it would be expected more papers to be included in the review.

-            Why limit to a single microcontroller (excluding popularity)? If cost, there are other microcontrollers available and devices like Arduinos that are used for prototyping, not large scale manufacturing

-            Benefits of the sensor is suggested, but not how, e.g., caregivers can recognize important changes. What changes? 

- Points are made, but specific support is needed, especially to the papers that you determined from your search of the literature and have included via inclusion and exclusion criteria.

-            Importantly, low-cost and costly are used through the paper, without quantification. In fact low-cost devices is one of the needs highlighted for research in this area. But simple searches on Google indicate that publicly available fully developed and ready for use oximeters can be got at similar prices (or less) to getting an Arduino and breakout board.

-            What are you aiming to convey in your results section? You primarily bring in the 4 papers in the discussion section

-            Why is reference 32 used? It does not seem to fit heart rate, respiratory, oximeter.

- Interesting points on challenges (for example) like additional sensors and advanced algorithms are just mentioned, without discussion of where the lack of accuracy arises and how these solutions would address the problems.

- Terms like traditional models need to be more clear. What specifically are you referring to.

 - Review misses detail of how accurate do these sensors need to be, what exact trends can the person at home look at for, 

- I believe the paper can be made much more concise.

- Also, your challenges are highlighted even before the methodology. It seems like the rest of the paper is redundant then.

- One overall suggestion is to be very clear on the focus of the paper, and ensure this is being conveyed to the reader, and that the appropriate information is being extracted and points clearly conveyed without repetition and concisely.

- Importantly, beyond just a table, it really needs to be clear where you are consulting the papers from the review in your results and discussion. Details and comparsions are extracted from far more papers.

Comments on the Quality of English Language

Further proof reading is needed as well as conciseness.

Round 2

Reviewer 1 Report

Comments and Suggestions for Authors

1.- Each chronic disease mentioned (COPD, asthma, cystic fibrosis, pneumonia, etc.) needs references that support the necessity of using pulse oximeters. Please, include up-to-date references for each disease to substantiate the discussion on the importance of pulse oximetry. These references should be from reputable sources and relevant studies that highlight the significance of oxygen monitoring in managing these conditions.

2.- The references to tables and figures in the text are not appropriately formatted. Ensure that every mention of a figure or table in the text is explicitly referenced, using phrases like “as shown in Figure 1,” to guide the reader clearly. This aligns with standard academic practices for clear and precise citation of figures.

3.- The manuscript contains repetitive phrases, such as “On the other hand,” and the overall language quality needs improvement. Review and refine the manuscript to improve the English quality. Use a variety of phrases such as “Conversely,” “Moreover,” or “Additionally” to avoid repetition. Ensure the language is formal, clear, and suitable for a high-impact journal. Consider a professional language editing service if necessary.

4.- The introduction does not include a summary of the manuscript’s structure. At the end of the introduction, include a brief overview of the manuscript’s structure, explaining what each section will cover. This helps readers navigate the document and understand the flow of information.

5.- Pages 6-10 and 13-16 contain too much text, which may cause readers to lose interest. Please, add figures or diagrams to break up the text and visually summarize key points. For instance, include a diagram summarizing the main challenges related to the implementation of low-cost pulse oximeters. In Section 4, add figures representing each low-cost pulse oximeter discussed, citing the author appropriately. Additionally, include a diagram at the end of Section 5 that encapsulates the critical factors to consider in relation to low-cost pulse oximeters in home care programs.

6.- The revised manuscript does not mention the specific light frequencies used to estimate oxygen saturation and heart rate. Include a detailed description of the light frequencies (e.g., 660 nm for red light and 940 nm for infrared light) used in pulse oximeters and explain how these are utilized to measure oxygen saturation and heart rate. This technical detail is crucial for a thorough understanding of the device's operation.

7.- Multiple tables (Tables 2, 3, 4, and 5) could be consolidated into a single table. Merge the tables into one comprehensive table, formatted according to the journal’s guidelines. This table should allow for easy comparison of the different pulse oximeters discussed, including columns for the author, device name, cost, accuracy, key features, and clinical applications.

8.- Explicitly state the unique contributions of this review to the field of low-cost pulse oximetry, such as identifying knowledge gaps, proposing future research directions, or highlighting under-explored areas. Consider adding a paragraph discussing possible future directions in research and development of low-cost oximeters.

9.- Incorporate more critical discussion of the studies reviewed, evaluating their methodologies, strengths, weaknesses, and relevance to the overall topic. This will add depth to the review and demonstrate a thorough understanding of the field.

Comments on the Quality of English Language

Certain phrases, such as "On the other hand," are used frequently and could be replaced with alternatives like "Furthermore" or "In contrast" to add variety. A careful review to improve sentence structure, clarity, and language diversity would be beneficial. Improving these aspects will make the manuscript more polished and easier to read. If necessary, you might consider using a professional editing service to help elevate the language quality.

Reviewer 2 Report

Comments and Suggestions for Authors

Dear Authors,

thank you for making adjustments based on previous recommendations. Your article "Advances and Challenges Associated with Low-Cost Pulse Oximeters in Home Care Programs: A Review" has made significant progress and many of the issues identified in the first review have been successfully addressed. However, I would like to highlight a few remaining areas that require further attention:

Some figure and table labels are still very general or non-descriptive. For example, the description "Table 6. Comparison of Low-Cost and Commercial Oximeters" does not correspond to the content of the table, which rather compares measurement principles. I recommend editing the captions throughout the article so that they more clearly reflect the content and purpose of individual tables and figures.

Diagram Figure 2: This diagram is a bit confusing. For example, the LED driver does not seem to provide data to the MCU, but vice versa, and it is also not clear why the LED driver is connected to "Signal Acquisition". I recommend re-examining this diagram and modifying it to better reflect the actual structure of the system and the correct directions of information flow.

Diagram Figure 3: The approach chosen for this diagram seems somewhat unprofessional, almost "childish". I think it would be more appropriate to use a technical and precise way of presentation, which would correspond better with the scientific focus of the article.

Check the references, some have very little information filled in.

At the end of the article, it should be emphasized that this method of solution is more suitable for emergency situations or for educational purposes. I can't imagine anyone monitoring their health with a sensor connected to an Arduino as presented. Your approach could be to design a simple kit that, if followed properly and tested (e.g. compared to commercial instruments), could be distributed for at least basic measurements.

Overall, I rate your work positively and after these minor adjustments I would recommend the article for acceptance.
